# Modulation of dorsal premotor cortex differentially influences I-wave excitability in primary motor cortex of young and older adults

Wei-Yeh Liao[1] , George M. Opie[1] , Ulf Ziemann[2,3] and John G. Semmler[1] 

[1] *Discipline of Physiology, School of Biomedicine, The University of Adelaide, Adelaide, Australia*
[2] *Department of Neurology & Stroke, Eberhard Karls University of Tübingen, Tübingen, Germany*
[3] *Hertie-Institute for Clinical Brain Research, Eberhard Karls University of Tübingen, Tübingen, Germany*

Handling Editors: Richard Carson & Charlotte Stagg

The peer review history is available in the Supporting information section of this article (https://doi.org/10.1113/JP284204#support-information-section).

**Abstract**  Previous research using transcranial magnetic stimulation (TMS) has demonstrated weakened connectivity between dorsal premotor cortex (PMd) and motor cortex (M1) with age. While this alteration is probably mediated by changes in the communication between the two regions, the effect of age on the influence of PMd on specific indirect (I) wave circuits within M1 remains unclear. The present study therefore investigated the influence of PMd on early and late

The Journal of Physiology

I-wave excitability in M1 of young and older adults. Twenty-two young (mean ± SD, 22.9 ± 2.9 years) and 20 older (66.6 ± 4.2 years) adults participated in two experimental sessions involving either intermittent theta burst stimulation (iTBS) or sham stimulation over PMd. Changes within M1 following the intervention were assessed with motor-evoked potentials (MEPs) recorded from the right first dorsal interosseous muscle. We applied posterior–anterior (PA) and anterior–posterior (AP) current single-pulse TMS to assess corticospinal excitability ($PA_{1mV}$; $AP_{1mV}$; $PA_{0.5mV}$, early; $AP_{0.5mV}$, late), and paired-pulse TMS short intracortical facilitation for I-wave excitability (PA SICF, early; AP SICF, late). Although PMd iTBS potentiated $PA_{1mV}$ and $AP_{1mV}$ MEPs in both age groups (both $P < 0.05$), the time course of this effect was delayed for $AP_{1mV}$ in older adults ($P = 0.001$). Furthermore, while $AP_{0.5mV}$, PA SICF and AP SICF were potentiated in both groups (all $P < 0.05$), potentiation of $PA_{0.5mV}$ was only apparent in young adults ($P < 0.0001$). While PMd influences early and late I-wave excitability in young adults, direct PMd modulation of the early circuits is specifically reduced in older adults.

(Received 1 December 2022; accepted after revision 12 May 2023; first published online 16 May 2023)

**Corresponding author** W.-Y. Liao: School of Biomedicine, The University of Adelaide, Adelaide, South Australia 5005, Australia. Email: wei-yeh.liao@adelaide.edu.au

**Abstract figure legend** We investigated the effects of ageing on the influence of dorsal premotor cortex (PMd) on indirect (I) wave circuits within motor cortex (M1) by applying intermittent theta burst stimulation (iTBS) to PMd in young and older adults. Transcranial magnetic stimulation (TMS) with posterior–anterior (PA) current to M1 was used to assess early I-wave circuits, whereas anterior–posterior (AP) current TMS was used to assess late I-wave circuits. Although we found that PMd iTBS increased M1 excitability for PA and AP TMS measures in young adults, facilitation in PA TMS was absent in older adults, suggesting that the influence of PMd on early I-waves is weakened with advancing age.

## Key points

- Interneuronal circuits responsible for late I-waves within primary motor cortex (M1) mediate projections from dorsal premotor cortex (PMd), but this communication probably changes with advancing age.
- We investigated the effects of intermittent theta burst stimulation (iTBS) to PMd on transcranial magnetic stimulation (TMS) measures of M1 excitability in young and older adults.
- We found that PMd iTBS facilitated M1 excitability assessed with posterior–anterior (PA, early I-waves) and anterior–posterior (AP, late I-waves) current TMS in young adults, with a stronger effect for AP TMS.
- M1 excitability assessed with AP TMS also increased in older adults following PMd iTBS, but there was no facilitation for PA TMS responses.
- We conclude that changes in M1 excitability following PMd iTBS are specifically reduced for the early I-waves in older adults, which could be a potential target for interventions that enhance cortical excitability in older adults.

## Introduction

While alterations to motor function are a universal effect of ageing, there is substantial variability in the way each person experiences these changes (Santoni et al., 2015). Whereas some older adults retain remarkable motor skills, others experience severe motor decline in the forms of reduced movement coordination, movement slowing

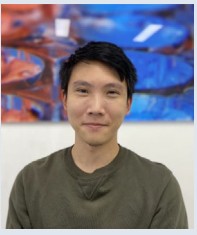

**Wei-Yeh Liao** studies how the ageing process modifies motor cortical plasticity and motor behaviour, and how motor function can be improved in older adults. In particular, his research at the University of Adelaide is focusing on using non-invasive brain stimulation to understand the role of dorsal premotor cortex on motor cortical function and how this contributes to movement deficits in older adults. He is also interested in the role of cerebellum on motor cortical plasticity and function.

and increased movement variability (Santoni et al., 2015; Seidler et al., 2010), all of which limit the ability of older adults to perform essential activities of daily life. Although structural changes in the motor system are important contributors to a decline in motor function with advancing age (Seidler et al., 2010), functional changes are also likely to play a major role. For example, the ageing motor system is associated with altered corticospinal (Bhandari et al., 2016) and intracortical excitability (Opie et al., 2018; Opie et al., 2020), in addition to weaker connectivity between nodes of the motor network (Green et al., 2018; Ni et al., 2015). These changes are likely to influence the potential for neuroplastic change in the motor system (Freitas et al., 2013; Semmler et al., 2021; Zimerman & Hummel, 2010), which is known to be an important neural substrate for motor behaviour and learning (Sanes & Donoghue, 2000). However, the neurophysiological mechanisms underpinning these changes with advancing age remain unclear.

Transcranial magnetic stimulation (TMS) is a type of non-invasive brain stimulation (NIBS) that can provide information on the physiology of specific neuronal networks within the motor system. Application of TMS over the motor cortex (M1) produces a complex descending volley within corticospinal neurons, which summate at the spinal cord to produce a motor-evoked potential (MEP) in targeted muscles (Di Lazzaro et al., 1998; Rossini et al., 2015). The first of these waves probably reflect direct activation of corticospinal neurons, whereas subsequent waves are thought to reflect indirect activation of distinct local intracortical networks (Di Lazzaro et al., 2012; Ziemann, 2020). The activities of these intracortical circuits, called indirect (I) waves, are generally referred to as early ($I_1$) or late ($I_2$, $I_3$) based on their order of appearance, and occur with a periodicity of ∼1.5 ms (Di Lazzaro et al., 2012; Ziemann, 2020). Early and late I-waves can be preferentially recruited by applying low-intensity single-pulse TMS with different current directions (Di Lazzaro et al., 2001; Ni et al., 2010; Sakai et al., 1997). For example, a posterior-to-anterior (PA) current (relative to the central sulcus) preferentially recruits early I-waves, whereas an anterior-to-posterior (AP) current preferentially recruits late I-waves (Di Lazzaro et al., 2001; Ni et al., 2010; Sakai et al., 1997). Using these measures, it has been reported that the ability of TMS to recruit late I-waves predicts the neuroplastic response to plasticity-inducing NIBS paradigms applied over M1 (Hamada et al., 2013; Wiethoff et al., 2014) and that the late I-waves are suggested to be behaviourally relevant for the acquisition of fine motor skills (Hamada et al., 2014).

In addition to MEPs recruited with different current directions, the excitability and temporal characteristics of the I-wave circuits can also be assessed using a paired-pulse TMS protocol referred to as short intracortical facilitation (SICF) (Tokimura et al., 1996; Ziemann et al., 1998). This protocol combines two perithreshold TMS pulses at short interstimulus intervals (ISIs), which reveal peaks of MEP facilitation at regular intervals that approximate the I-wave periodicity, and is thought to result from a facilitatory interaction between the early and late I-waves generated by each TMS pulse (Ziemann et al., 1998). Importantly, we have previously shown that there is a decreased excitability of all SICF peaks in older adults (Opie et al., 2018), with temporal delays in the late SICF peaks (that are thought to reflect activity of late $I_3$ waves), which play a role in NIBS-induced plasticity (Opie et al., 2018) and are predictive of motor behaviour in older adults (Opie et al., 2020).

Although changes in the late I-wave circuits may be critical for understanding age-related differences in M1 excitability and motor behaviour, it is unclear what mechanisms are mediating the alterations in these circuits. I-wave circuits seem to represent a point of convergence for inputs from other nodes of the motor system such as dorsal premotor cortex (PMd) (Groppa et al., 2012). The PMd plans, predicts and corrects movements during motor learning by updating the activity of M1 (Chouinard et al., 2005; Nowak et al., 2009; Parikh & Santello, 2017). Recent studies have reported that modulation of PMd excitability using repetitive TMS (rTMS) techniques such as theta burst stimulation (TBS) is able to modify M1 excitability (Meng et al., 2020), as well as alter the neuroplastic response of M1 and influence motor skill learning (Huang et al., 2018). The ability to recruit late I-waves with TMS has also been shown to predict the strength of PMd–M1 connectivity (Volz et al., 2015), in addition to the neuroplastic response to M1 TMS interventions (Hamada et al., 2013; Volz et al., 2019). Furthermore, PMd–M1 connectivity has been shown to decline with age (Ni et al., 2015). However, the influence of PMd on early and late I-wave circuits in M1, and how these change with age, are not known.

Therefore, the present study aimed to investigate the influence of PMd on I-wave excitability in M1 of young and older adults. Intermittent TBS (iTBS) was used to upregulate PMd excitability in young and older participants, and different I-wave circuits were assessed by varying the direction of current used to apply TMS over M1. Although we expected iTBS over PMd to selectively modulate late I-wave activity, we hypothesised that the neuroplastic response of the late circuits to PMd iTBS would be weaker in older adults, given the probable alterations in late I-wave activity and PMd–M1 connectivity with advancing age.

## Methods

Twenty-two young (mean ± standard deviation, 22.9 ± 2.9 years; range, 18−29 years; females = 13) and 20 older (66.6 ± 4.2 years; 60−76 years; females = 11) adults were recruited for the present study via advertisements placed on notice boards within The University of Adelaide and the wider community, in addition to social media platforms. Exclusion criteria included a history of psychiatric or neurological disease, current use of medications that affect the CNS or left handedness. Suitability for TMS was assessed using a standard screening questionnaire (Rossi et al., 2011). The experiment was conducted in accordance with the *Declaration of Helsinki*, and was approved by The University of Adelaide Human Research Ethics Committee (H-026-2008). Participants provided written, informed consent prior to participation.

### Experimental arrangement

All participants attended a PMd iTBS and PMd sham iTBS session (Fig. 1*A*), with a washout period of at least 1 week between sessions. The same experimental protocol was used in both sessions, with the order of intervention randomised between participants (Fig. 1*B*). As diurnal variations in cortisol are known to influence the neuroplastic response to TMS (Sale et al., 2008), all sessions were completed between 11.00 and 17.00 h at approximately the same time of day within each participant.

During each experimental session, participants were seated in a comfortable chair with their hands resting and relaxed. Surface EMG was recorded from the first dorsal interosseous (FDI) of the right hand using two Ag-AgCl electrodes arranged in a belly-tendon montage on the skin overlying the muscle, with a third electrode attached above the styloid process of the right ulnar used to ground the electrodes. EMG signals were amplified (300×) and filtered (band-pass 20 Hz to 1 kHz) using a CED 1902 signal conditioner (Cambridge Electronic Design, Cambridge, UK) before being digitised at 2 kHz using a CED 1401 analog-to-digital converter. Signal noise associated with mains power was removed using a Humbug mains noise eliminator (Quest Scientific, North Vancouver, Canada). EMG signals were stored on a PC for offline analysis. Real-time EMG signals were displayed on an oscilloscope placed in front of the participant to facilitate muscle relaxation during the experiment. During breaks throughout each experimental session, participants remained seated and were instructed to maintain relaxation of the right hand, but were allowed access to mobile devices or books with their left hand.

### Experimental procedures

**Transcranial magnetic stimulation (TMS).** A branding iron coil (70 mm diameter) connected to two Magstim $200^2$ magnetic stimulators (Magstim, Whitland, UK) via a BiStim unit was used to apply TMS to left M1. The

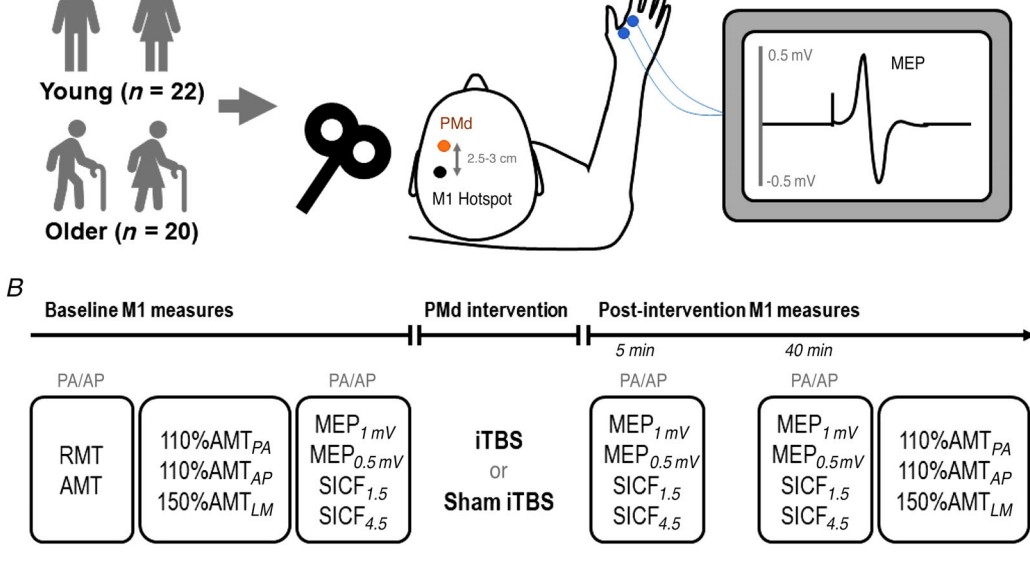

**Figure 1. Summary of experimental setup and procedure**
*A*, subject sample and experimental setup. *B*, experimental procedure. PA, posterior-to-anterior; AP, anterior-to-posterior; LM, lateral-to-medial; RMT, resting motor threshold; AMT, active motor threshold; $MEP_{1mV}$, standard MEP of ∼1 mV at baseline; $MEP_{0.5mV}$, MEP of ∼0.5 mV at baseline; SICF, short intracortical facilitation; PMd, dorsal premotor cortex; iTBS, intermittent theta burst stimulation. [Colour figure can be viewed at wileyonlinelibrary.com]

coil was held tangentially to the scalp at an angle of 45° to the sagittal plane, inducing a PA current relative to the central sulcus. The M1 hotspot was identified as the location producing the largest and most consistent MEPs within the relaxed FDI muscle of the right hand (Rossini et al., 2015). This location was marked on the scalp for reference and continuously monitored throughout each experimental session. All baseline and post-intervention (5 min, 40 min) TMS pulses were applied at a rate of 0.2 Hz, with a 10% jitter between trials to avoid anticipation of the stimulus.

Resting motor threshold (RMT) over M1 was recorded at the beginning of each experimental session as the lowest stimulus intensity (expressed as a percentage of maximum stimulator output; %MSO) producing an MEP amplitude $\geq 50\ \mu$V in at least 5 out of 10 trials during relaxation of the right FDI (Rossini et al., 2015). We then asked the participants to activate and maintain an $\sim$10% contraction of the right FDI during the assessment of active motor threshold (AMT), defined as the lowest %MSO producing an MEP amplitude $\geq 200\ \mu$V in at least 5 out of 10 trials during concurrent low-level activation of the muscle (D'Ostilio et al., 2016; Hamada et al., 2013). These measures were then repeated using the AP current by rotating the coil 180°. Following AMT, the stimulus intensities producing a standard MEP amplitude approximating 1 mV ($\text{MEP}_{1mV}$; $\text{PA}_{1mV}$ and $\text{AP}_{1mV}$), in addition to an MEP amplitude approximating 0.5 mV ($\text{MEP}_{0.5mV}$; $\text{PA}_{0.5mV}$ and $\text{AP}_{0.5mV}$), when averaged over 20 trials, were identified. The stimulation intensities were adjusted and the trials were re-recorded if the average MEP amplitude did not fall in the range of 0.8–1.2 mV for $\text{MEP}_{1mV}$ and 0.4–0.6 mV for $\text{MEP}_{0.5mV}$. The same intensities were then applied following the intervention (PMd iTBS) to assess changes in corticospinal excitability.

**I-wave recruitment.** To investigate the ability to recruit I-waves, onset latencies were recorded for MEPs produced by application of PA and AP current TMS, and expressed relative to the onset of responses generated by direct activation of corticospinal neurons using lateral-to-medial (LM) current TMS (Hamada et al., 2013). A block of 15 MEP trials in the active FDI was recorded for 110% $\text{AMT}_{PA}$ and $\text{AMT}_{AP}$, in addition to 150% $\text{AMT}_{LM}$. If 150% $\text{AMT}_{LM}$ exceeded 100% MSO, 100% MSO was used, or if 150% $\text{AMT}_{LM}$ was below 50% MSO, 50% MSO was used (Hamada et al., 2013). The differences in mean onset latencies between PA and LM (PA-LM) and AP and LM (AP-LM) MEPs recorded in the active muscle were calculated as measures of early and late I-wave recruitment efficiency (Hamada et al., 2013). In an attempt to reduce the confounding influence of muscle contraction (required to quantify I-wave recruitment) on neuroplasticity induction (Goldsworthy et al., 2015;

Huang et al., 2008; Thirugnanasambandam et al., 2011), these measures were recorded at the start and at the end of the experimental session, at least 1 h apart from the PMd iTBS.

**I-wave excitability.** The paired-pulse TMS SICF was used to index early and late I-wave excitability. SICF utilised a conditioning stimulus set at 90% RMT following a test stimulus set at $\text{MEP}_{1mV}$ (Ziemann et al., 1998), and was delivered using PA and AP current directions, which have been suggested to target the activity of different early and late I-wave circuits (Opie et al., 2021). An ISI of 1.5 ms was used to assess early I-waves (PA $\text{SICF}_{1.5}$, AP $\text{SICF}_{1.5}$) while 4.5 ms was used to assess late I-waves (PA $\text{SICF}_{4.5}$, AP $\text{SICF}_{4.5}$) (Ziemann et al., 1998). Each PA and AP SICF block performed at baseline and after intervention consisted of 36 stimuli: 12 single-pulse-only trials and 12 paired-pulse trials for each ISI.

**Theta burst stimulation (TBS).** iTBS was delivered over left PMd using a Magstim Super-rapid stimulator (Magstim), connected to a figure-of-eight coil (70 mm in diameter). The coil was held tangentially to the scalp, at an angle of 45° to the sagittal plane, with the handle pointing backwards and laterally, inducing a biphasic pulse with an initial PA current followed by an AP return current (Suppa et al., 2008). In accordance with existing literature, iTBS consisted of bursts of three pulses given at a frequency of 50 Hz. Each triplet was repeated 10 times at 200 ms within a 2 s train, and this was repeated every 8 s for 20 cycles, totalling 600 pulses (Huang et al., 2005, 2008, 2018; Meng et al., 2020). The location of left PMd was defined as 8% of the distance between the nasion and inion ($\sim$2.5–3 cm) anterior to the M1 hotspot, consistent with previous work (Huang et al., 2018; Koch et al., 2007; Meng et al., 2020; Münchau et al., 2002). Both the M1 hotspot and left PMd location were digitally recorded relative to the standard MNI-ICBM152 template using Brainsight neuronavigation (Rogue Research, Montreal, Quebec, Canada). The digital recordings were then used to guide the assessment of AMT over M1 with the Magstim Super-rapid stimulator, in addition to maintaining consistent coil positioning during the application of iTBS over left PMd at 80% AMT ($\text{AMT}_{Rapid}$).

In contrast, sham iTBS was delivered using a sham figure-of-eight coil (70 mm in diameter) to replicate the pulse noise, while a bar electrode connected to a constant current stimulator (Digitimer, Welwyn Garden City, UK) concurrently applied electrical stimuli (1.5 mA intensity) to the scalp above left PMd to mimic the pulse sensation. Visual analogue scales (VAS) were used following each intervention to assess the degree of discomfort, FDI activation and localisation of scalp sensation associated with TBS.

## Data analysis

Visual inspection of EMG data was completed offline, and any trials with EMG activity exceeding 25 $\mu$V in the 100 ms prior to stimulus application in the resting muscle were excluded from analysis (~2.5% of MEP trials removed). The amplitude of MEPs obtained from recordings in the resting muscle was measured peak-to-peak and expressed in millivolts. The MEP onset latency obtained from recordings in the active muscle was assessed with a semi-automated process using a custom script within the Signal program (v 6.02, Cambridge Electronic Design) and expressed in milliseconds. Onset of MEPs for each trial from active muscle recordings was defined as the point at which the rectified EMG signal following the stimulus artefact exceeded the mean EMG amplitude plus 2 SD within the 100 ms pre-stimulus. Within each participant, the mean LM MEP latencies were subtracted from the mean PA and AP MEP latencies to determine PA-LM and AP-LM MEP latency differences of the means. For baseline measures of SICF, individual paired-pulse MEP amplitude was expressed as a percentage of the mean MEP amplitude produced by single-pulse TMS in the same block. For all post-iTBS measures of SICF, individual MEP amplitudes produced by paired-pulse TMS were expressed as a percentage of the mean MEP amplitude produced by single-pulse TMS recorded at baseline, as undertaken previously (Cash et al., 2009; Liao et al., 2022; Opie et al., 2021). This was performed because the increase in post-intervention single-pulse MEP amplitude is correlated with the increase in post-intervention SICF, and normalising to the post-intervention test MEP amplitude underestimates the change in excitability of the I-wave generating networks (Cash et al., 2009). We therefore also tested this relationship in the present study using Spearman's rank-order correlation analysis, and found that post-iTBS increases in single-pulse MEP amplitude were related to increases in paired-pulse MEP amplitude ($\rho = 0.8$, $P < 0.0001$; see Results). For all post-intervention TMS measures (MEP amplitude and onset latency), effects of PMd iTBS were quantified by expressing the post-intervention responses as a percentage of the baseline responses.

## Statistical analysis

Visual inspection and Kolmogorov–Smirnov tests of the data residuals revealed non-normal, positively skewed distributions for all TMS data. Consequently, generalised linear mixed models (GLMMs), which can account for non-normal distributions (Lo & Andrews, 2015; Puri & Hinder, 2022), were used to perform all statistical analyses. Each model assessing MEP amplitude included single trial data with repeated measures and was fitted with gamma distributions (Lo & Andrews, 2015), with all random subject effects included (intercepts and slopes) (Barr et al., 2013). Identity link functions were used for raw MEP amplitudes while log link functions were used for responses expressed as a percentage (baseline-normalised responses and baseline SICF) (Lo & Andrews, 2015; Puri & Hinder, 2022). To optimise model fit, we tested different covariance structures, and the structure providing the best fit (assessed with the Bayesian Schwartz Criterion; BIC) within a model that was able to converge was used in the final model. Two-factor GLMMs were used to compare the effects of session (iTBS, sham) and age (young, older) on stimulator output intensities, investigated for PA and AP orientations of RMT, AMT, MEP$_{1mV}$ and MEP$_{0.5mV}$, in addition to AMT$_{LM}$ and AMT$_{Rapid}$. Two-factor GLMMs were used to compare effects of session and age at baseline for measures of corticospinal excitability (PA$_{1mV}$, AP$_{1mV}$, PA$_{0.5mV}$, AP$_{0.5mV}$) and intracortical excitability (PA SICF$_{1.5}$, PA SICF$_{4.5}$, AP SICF$_{1.5}$, AP SICF$_{4.5}$). In contrast, a three-factor model including mean MEP latency data (PA-LM, AP-LM) was used to compare the effects of session, age and orientation (PA, AP) on I-wave recruitment at baseline.

Changes in MEP$_{1mV}$ measures of corticospinal excitability following the intervention were investigated by assessing effects of session, time (5 min, 40 min) and age in two separate models for baseline-normalised PA$_{1mV}$ and AP$_{1mV}$ MEP amplitudes. Changes in MEP$_{0.5mV}$ measures of corticospinal excitability following the intervention were investigated by assessing effects of session, time and age in two separate models for baseline-normalised PA$_{0.5mV}$ and AP$_{0.5mV}$ MEP amplitudes. Changes in SICF measures of I-wave excitability following the intervention were investigated by assessing effects of session, time and age in four separate models for baseline-normalised PA SICF$_{1.5}$, PA SICF$_{4.5}$, AP SICF$_{1.5}$ and AP SICF$_{4.5}$ responses. Changes in I-wave recruitment following the intervention were investigated by assessing effects of session, age and coil orientation on baseline-normalised average PA-LM and AP-LM MEP latencies. For all models, investigations of main effects and interactions were performed using custom contrasts with Bonferroni correction, and significance was set at $P < 0.05$. Data for all models are presented as estimated marginal means (EMMs) and 95% confidence intervals (95% CI), whereas pairwise comparisons are presented as the estimated mean difference (EMD) and 95% CI for the estimate.

Furthermore, we used Spearman's rank-order correlation to assess the relationship between different variables. Specifically, the mean baseline MEP onset latencies were correlated against changes in cortico-spinal and intracortical excitability, in addition to changes in I-wave recruitment following the intervention. Furthermore, the individual age of subjects was correlated against changes in corticospinal and

**Table 1. Baseline TMS intensities between sessions for young and older adults**

| Measure | Young | | Older | |
| --- | --- | --- | --- | --- |
| | iTBS | Sham | iTBS | Sham |
| **PA** | | | | |
| $RMT_{PA}$ (%MSO) | 48.7 [44.4, 53.1] | 49.0 [44.7, 53.3] | 53.0 [48.5, 57.5] | 53.4 [48.8, 57.9] |
| $AMT_{PA}$ (%MSO) | 41.8 [38.1, 45.5] | 41.8 [38.1, 45.5] | 44.4 [40.5, 48.3] | 45.4 [41.5, 49.2] |
| $PA_{1mV}$ (%MSO) | 58.5 [52.6, 64.4] | 58.9 [53.0, 64.8] | 71.3 [64.9, 77.7][a] | 69.1 [62.7, 75.5][a] |
| $PA_{0.5mV}$ (%MSO) | 54.9 [49.3, 60.5] | 54.8 [49.1, 60.4] | 64.5 [58.5, 70.6][a] | 64.0 [57.9, 70.0][a] |
| **AP** | | | | |
| $RMT_{AP}$ (%MSO) | 60.7 [56.0, 65.4] | 61.9 [57.2, 66.6] | 63.4 [58.3, 68.5] | 65.1 [60.0, 70.1] |
| $AMT_{AP}$ (%MSO) | 53.9 [49.1, 58.7] | 54.6 [49.8, 59.5] | 57.2 [52.1, 62.3] | 58.9 [53.8, 63.9] |
| $AP_{1mV}$ (%MSO) | 74.0 [67.5, 80.4] | 74.6 [68.1, 81.0] | 86.4 [79.0, 93.8] | 81.8 [74.3, 89.2] |
| $AP_{0.5mV}$ (%MSO) | 70.0 [64.5, 75.5] | 70.0 [64.4, 75.5] | 76.5 [70.2, 82.8] | 75.0 [68.7, 81.3] |
| **LM** | | | | |
| $AMT_{LM}$ (%MSO) | 46.5 [42.4, 50.6] | 48.1 [44.0, 52.3] | 49.7 [45.4, 54.1] | 48.1 [43.8, 52.4] |
| **TBS** | | | | |
| $AMT_{Rapid}$ (%MSO) | 52.5 [48.5, 56.5] | 53.6 [49.6, 57.6] | 53.3 [49.1, 57.4] | 55.0 [50.8, 59.1] |

Data show EMM [95% CI; lower, upper].
[a] $P < 0.05$ compared to the same session in young.

intracortical excitability, and I-wave recruitment following the intervention to identify if changes in excitability and recruitment were driven by age-related effects. Correlations are presented as Spearman's $\rho$ with false discovery rate-adjusted $P$-value of 0.05 following the Benjamini–Hochberg procedure. Lastly, differences in the perception of iTBS and sham iTBS were investigated by comparing VAS responses using paired $t$ tests with Bonferroni correction ($P < 0.0167$), with data presented as mean $\pm$ SD.

## Results

All participants completed the two experimental sessions without adverse reactions. We were unable to record $PA_{1mV}$ (and PA SICF) in one older male participant, $AP_{0.5mV}$ in five participants (one young female; two older females, two older males) and $AP_{1mV}$ (and AP SICF) in six participants (one young female; three older females, two older males) due to high thresholds of activation (mean $RMT_{PA} = 76.0\%$ MSO, mean $RMT_{AP} = 84.7\%$ MSO). MEP latency data of one young participant were excluded due to contamination by stimulation artefacts. Baseline TMS intensities are presented in Table 1. Stimulation intensities for $PA_{1mV}$ varied between age groups ($F_{1,78} = 9.19$, $P = 0.003$), with *post hoc* tests revealing higher intensities for older adults relative to young adults (EMD = 11.5% [3.9, 19.0], $P = 0.003$). Similarly, intensities for $PA_{0.5mV}$ were different between age groups ($F_{1,80} = 5.54$, $P = 0.021$), with *post hoc* comparisons revealing higher intensities for older adults relative to young adults (EMD = 9.4% [1.5, 17.4],

$P = 0.021$). There were no main effects or interactions for all other baseline stimulation intensities (all $P > 0.05$).

Baseline MEP amplitudes for the assessment of corticospinal and intracortical excitability, in addition to MEP latencies, are shown in Table 2. For AP $SICF_{1.5}$ responses, there was an interaction between session and age group ($F_{1,831} = 5.48$, $P = 0.020$). *Post hoc* comparisons revealed higher MEP amplitudes in young adults during the iTBS session compared to sham (EMD = 34.8% [3.8, 65.7], $P = 0.028$), in addition to the MEP amplitudes of older adults in the iTBS session (EMD = 44.1% [1.0, 87.2], $P = 0.045$). In addition, baseline MEP latencies differed between coil orientations ($F_{1,156} = 247.41$, $P < 0.0001$), with shorter PA-LM latencies compared to AP-LM latencies (EMD = 1.9 ms [1.6, 2.1], $P < 0.0001$). There was also an interaction between coil orientation and age groups ($F_{1,156} = 3.92$, $P = 0.049$), with *post hoc* tests revealing shorter PA-LM latencies compared to AP-LM latencies in both young (EMD = 2.1 ms [1.8, 2.4], $P < 0.0001$) and older adults (EMD = 1.6 ms [1.3, 2.0], $P < 0.0001$). PA-LM latencies were also longer in older adults compared to young adults (EMD = 0.4 ms [0.0, 0.8], $P = 0.042$). There were no main effects or interactions for all other baseline responses (all $P > 0.05$).

### Changes in M1 excitability and I-wave recruitment after PMd iTBS

The participants' perceptions of the intervention are shown in Table 3. While there were no differences in the extent of discomfort ($t_{37} = 1.12$, $P = 0.272$) or FDI activation ($t_{37} = 1.76$, $P = 0.088$) experienced by the

**Table 2. Baseline responses of corticospinal and intracortical excitability and recruitment between sessions**

| Measure | | Young | | Older | |
|---|---|---|---|---|---|
| | | iTBS | Sham | iTBS | Sham |
| **PA** | | | | | |
| $PA_{1mV}$ (mV) | | 0.95 [0.87, 1.04] | 0.90 [0.82, 1.00] | 0.86 [0.78, 0.95] | 0.87 [0.78, 0.96] |
| $PA_{0.5mV}$ (mV) | | 0.52 [0.46, 0.57] | 0.51 [0.45, 0.57] | 0.48 [0.42, 0.54] | 0.43 [0.37, 0.48] |
| PA SICF (%test) | 1.5 ms | 186.1 [150.5, 230.1] | 198.6 [160.6, 245.6] | 155.4 [123.7, 195.2] | 174.9 [139.2, 219.8] |
| | 4.5 ms | 124.4 [108.2, 143.0] | 119.3 [103.7, 137.2] | 131.2 [112.8, 152.6] | 140.5 [120.8, 163.4] |
| PA-LM latency (ms) | | 1.56 [1.24, 1.88] | 1.48 [1.16, 1.80] | 1.73 [1.40, 2.07][a] | 2.12 [ 1.77, 2.46][a] |
| **AP** | | | | | |
| $AP_{1mV}$ (mV) | | 0.97 [0.87, 1.06] | 0.91 [0.81, 1.00] | 0.81 [0.70, 0.91] | 0.88 [0.77, 0.99] |
| $AP_{0.5mV}$ (mV) | | 0.49 [0.45, 0.54] | 0.49 [0.45, 0.54] | 0.46 [0.42, 0.51] | 0.49 [0.44, 0.53] |
| AP SICF (%test) | 1.5 ms | 178.3 [148.8, 213.5] | 143.5 [119.8, 172.0][c] | 134.2 [108.3, 166.2][a] | 153.4 [123.9, 190.0] |
| | 4.5 ms | 121.8 [108.8, 136.2] | 101.5 [90.6, 113.8] | 101.5 [88.7, 116.0] | 104.5 [91.6, 119.4] |
| AP-LM latency (ms) | | 3.56 [3.17, 3.94][b] | 3.67 [3.27, 4.06][b] | 3.46 [3.07, 3.85][b] | 3.64 [3.23, 4.04][b] |

Data show EMM [95% CI; lower, upper].
[a] $P < 0.05$ compared to the same session in young.
[b] $P < 0.05$ compared to the same session in PA-LM latency.
[c] $P < 0.05$ compared to iTBS session within same age group.

**Table 3. Comparison of VAS responses between sessions**

| Question | PMd iTBS | PMd Sham |
|---|---|---|
| How uncomfortable were the TMS pulses (0, not uncomfortable at all; 10, highly uncomfortable)? | 2.74 ± 2.60 | 2.16 ± 2.19 |
| If there were any twitches in the right hand, how strong were they (0, no twitches; 10, very strong cramp)? | 0.97 ± 1.62 | 0.53 ± 1.06 |
| How localised were the sensations from TMS pulses (0, highly localised; 10, widespread)? | 1.79 ± 1.79 | 0.76 ± 1.10[a] |

Data show mean ± SD.
[a] $P < 0.0167$ compared to iTBS.

participants between sessions, the locality of stimulation differed ($t_{37} = 3.83$, $P = 0.0005$), with the sensation of iTBS perceived as more widespread compared to sham.

### Changes in single-pulse TMS measures of corticospinal excitability

Changes in $MEP_{1mV}$ measures of corticospinal excitability after PMd iTBS are shown in Fig. 2. While MEP amplitudes for $PA_{1mV}$ did not vary between ages ($F_{1,3063} = 0.60$, $P = 0.439$) or time points ($F_{1,3063} = 0.10$, $P = 0.755$), they differed between sessions ($F_{1,3063} = 12.21$, $P = 0.0005$), with *post hoc* comparisons showing increased MEP amplitudes following PMd iTBS relative to sham (EMD = 32.5% [13.5, 51.6], $P = 0.001$; Fig. 2A). There were no interactions between factors (all $P > 0.05$; Fig. 2C presents changes in $PA_{1mV}$ between sessions for young and older adults over time).

$AP_{1mV}$ MEP amplitudes varied between sessions ($F_{1,2652} = 30.50$, $P < 0.0001$), with comparisons showing increased MEP amplitudes following PMd iTBS relative to sham (EMD = 43.6% [27.2, 59.9], $P < 0.0001$; Fig. 2B). $AP_{1mV}$ also differed between time points ($F_{1,2652} = 4.04$, $P = 0.044$), with comparisons revealing increased MEP amplitudes at 40 min relative to 5 min following intervention (EMD = 15.8% [0.3, 31.3], $P = 0.046$). There were no differences between age groups ($F_{1,2652} = 0.22$, $P = 0.636$) and no two-way interactions between factors (all $P > 0.05$). However, there was a three-way interaction between session, time and age ($F_{1,2652} = 4.59$, $P = 0.032$; Fig. 2D). *Post hoc* analysis shows increased MEP amplitudes in young adults following PMd iTBS compared to sham at 5 min (EMD = 58.3% [32.3, 84.4], $P < 0.0001$) and 40 min (EMD = 38.0% [11.1, 64.8], $P = 0.006$), while this increase was only apparent for older adults at 40 min (EMD = 54.3% [21.6, 87.0], $P = 0.001$). MEP amplitudes in older adults following iTBS were also

increased at 40 min relative to 5 min (EMD = 36.6% [2.8, 70.5], $P = 0.034$).

Changes in $MEP_{0.5mV}$ measures of corticospinal excitability are presented in Fig. 3. MEP amplitudes with $PA_{0.5mV}$ TMS did not differ between time points ($F_{1,3131} = 1.23$, $P = 0.268$) or age groups ($F_{1,3131} = 0.01$, $P = 0.920$), but varied between sessions ($F_{1,3131} = 17.22$, $P < 0.0001$), with responses following PMd iTBS increased compared to sham (EMD = 43.9% [22.1, 65.7], $P < 0.0001$). There was also an interaction between session and age ($F_{1,3131} = 4.55$, $P = 0.033$; Fig. 3A), with *post hoc* comparisons showing increased MEP amplitudes in young adults following PMd iTBS compared to sham (EMD = 67.4% [35.0, 99.8], $P < 0.0001$). There were no other interactions (all $P > 0.05$).

While MEP amplitudes for $AP_{0.5mV}$ did not vary between time points ($F_{1,2709} = 0.21$, $P = 0.645$) or age groups ($F_{1,2709} = 0.60$, $P = 0.441$), there was a difference between sessions ($F_{1,2709} = 28.18$, $P < 0.0001$; Fig. 3B), with responses following PMd iTBS increased relative to sham (EMD = 60.5% [36.5, 84.5], $P < 0.0001$). Additionally, there was also an interaction between session and age ($F_{1,2709} = 4.34$, $P = 0.037$), with *post hoc* comparisons showing increased MEP amplitudes following PMd iTBS relative to sham in young (EMD = 89.4% [53.9, 124.8], $P < 0.0001$) and older adults (EMD = 34.8% [2.0, 67.5], $P = 0.038$). There were no other interactions (all $P > 0.05$).

## Changes in paired-pulse TMS measures of I-wave excitability

Correlation analysis of post-intervention single- and paired-pulse responses revealed that increases in single-pulse test MEP amplitude were correlated with increases in paired-pulse MEP amplitude ($\rho = 0.8$, $P < 0.0001$). Changes in paired-pulse measures of SICF are presented in Fig. 4. Responses for PA $SICF_{1.5}$ did not differ between sessions ($F_{1,1900} = 1.17$, $P = 0.279$), time points ($F_{1,1900} = 1.74$, $P = 0.188$) or age groups ($F_{1,1900} = 0.07$, $P = 0.795$), but there was an interaction between session and time ($F_{1,1900} = 4.77$, $P = 0.029$; Fig. 4A). Comparisons showed increased PA $SICF_{1.5}$ at 40 min following PMd iTBS relative to sham (EMD = 15.3% [1.1, 29.5], $P = 0.035$), and compared to 5 min (EMD = 16.6% [2.4, 30.8], $P = 0.022$). PA $SICF_{4.5}$ responses (Fig. 4B) did not vary between sessions ($F_{1,1902} = 0.52$, $P = 0.469$), time points ($F_{1,1902} = 0.00$, $P = 1.00$) or age groups ($F_{1,1902} = 1.07$, $P = 0.302$), and there were no interactions between factors (all $P > 0.05$).

Responses for AP $SICF_{1.5}$ (Fig. 4C) did not vary between sessions ($F_{1,1636} = 2.51$, $P = 0.113$), time points ($F_{1,1636} = 0.64$, $P = 0.425$) or age groups ($F_{1,1636} = 0.21$, $P = 0.649$), and there were no interactions between factors (all $P > 0.05$). In contrast, while AP $SICF_{4.5}$ responses did not differ between time points ($F_{1,1636} = 0.13$, $P = 0.721$) or age groups ($F_{1,1636} = 0.65$, $P = 0.420$), they varied between sessions ($F_{1,1636} = 9.68$, $P = 0.002$), with *post hoc* comparisons revealing increased AP $SICF_{4.5}$ following PMd iTBS compared to sham (EMD = 30.7% [10.8, 50.6], $P = 0.002$). There was also an interaction between session and time ($F_{1,1636} = 5.18$, $P = 0.023$; Fig. 4D), with *post*

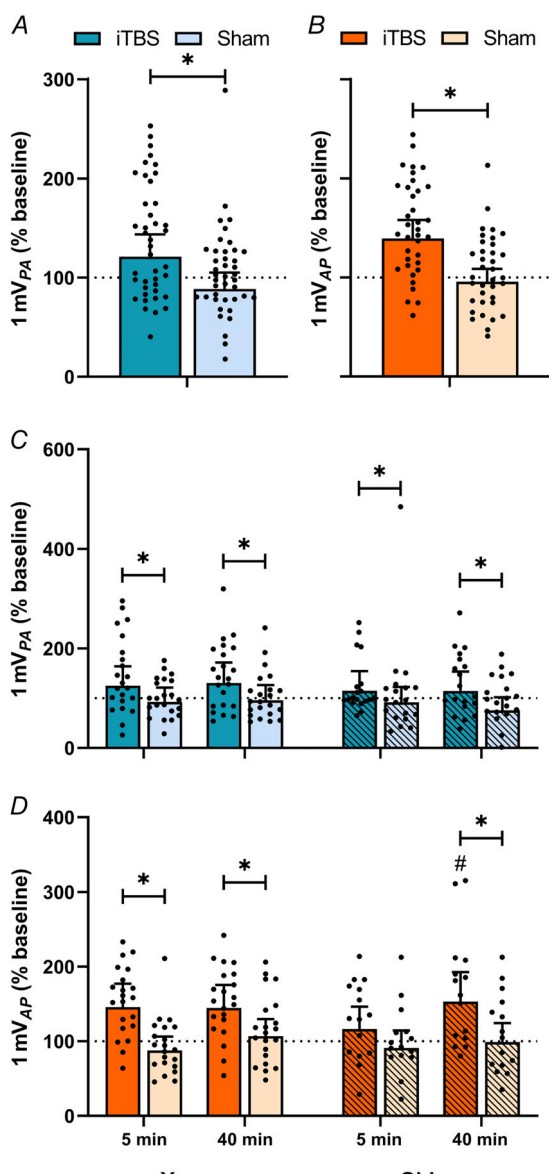

**Figure 2. Results for MEP$_{1mV}$**
*A* and *B*, changes in MEP$_{1mV}$ measures of corticospinal excitability (*A*, PA$_{1mV}$, blue; *B*, AP$_{1mV}$, orange) following PMd iTBS (darker hue) and sham (lighter hue) stimulation in all participants. *C* and *D*, changes in PA$_{1mV}$ (*C*) and AP$_{1mV}$ (*D*) in young (no stripes) and older (stripes) adults at 5 and 40 min following PMd iTBS and sham. Data show EMM [95% CI] with individual subject means. *$P < 0.05$. [Colour figure can be viewed at wileyonlinelibrary.com]

*hoc* comparisons showing increased AP SICF$_{4.5}$ at 40 min following PMd iTBS relative to sham (EMD = 46.6% [21.5, 71.8], $P = 0.0003$). There were no other interactions (all $P > 0.05$).

## Changes in PA and AP TMS latencies

Changes in MEP latency for PA and AP TMS after PMd iTBS are shown in Fig. 5. The change in MEP latency did not differ between sessions ($F_{1,156} = 1.30$, $P = 0.256$), coil orientations ($F_{1,156} = 0.03$, $P = 0.858$) or age groups

($F_{1,156} = 0.10$, $P = 0.758$), and there were no interactions between factors (all $P > 0.05$).

## Correlation analyses

Baseline PA-LM latencies did not predict changes in corticospinal excitability (PA$_{1mV}$, AP$_{1mV}$, PA$_{0.5mV}$, AP$_{0.5mV}$) or intracortical excitability (PA SICF$_{1.5}$, PA SICF$_{4.5}$, AP SICF$_{1.5}$, AP SICF$_{4.5}$, PA-LM latency, AP-LM latency; all $P > 0.05$). In contrast, baseline AP-LM latencies predicted changes in AP-LM latency following

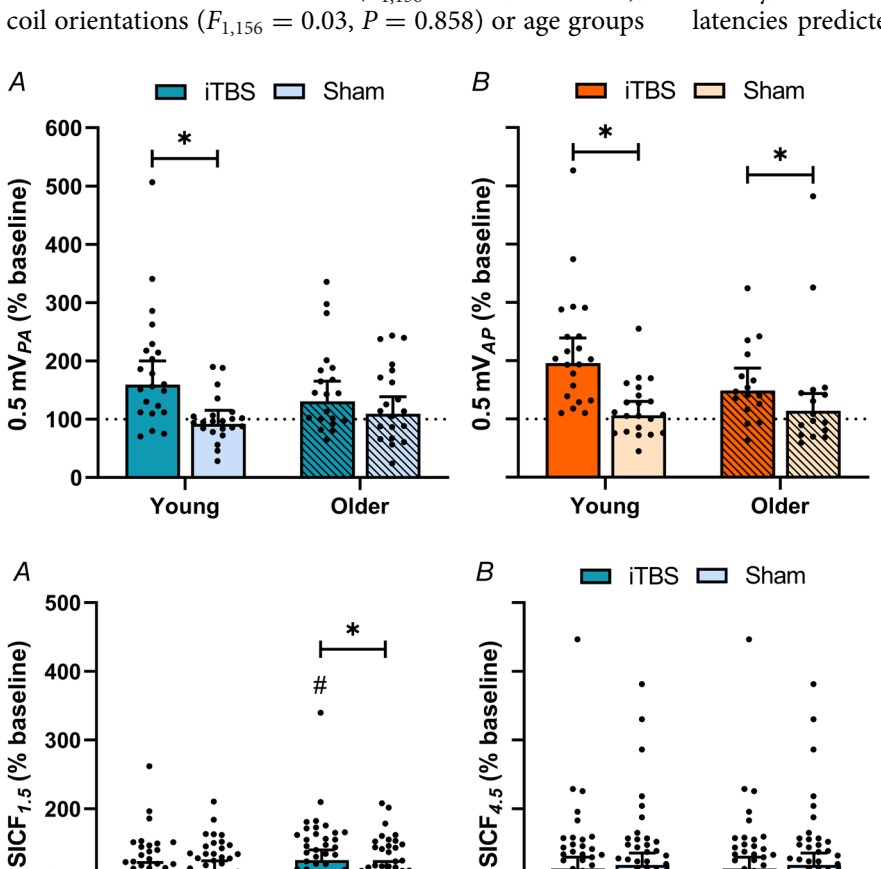

**Figure 3. Results for MEP$_{0.5mV}$**
*A* and *B*, changes in MEP$_{0.5mV}$ measures of corticospinal excitability (*A*, PA$_{0.5mV}$, blue; *B*, AP$_{0.5mV}$, orange) in young (no stripes) and older (stripes) adults following PMd iTBS (darker hue) and sham (lighter hue) stimulation. Data show EMM [95% CI] with individual subject means. *$P < 0.05$. [Colour figure can be viewed at wileyonlinelibrary.com]

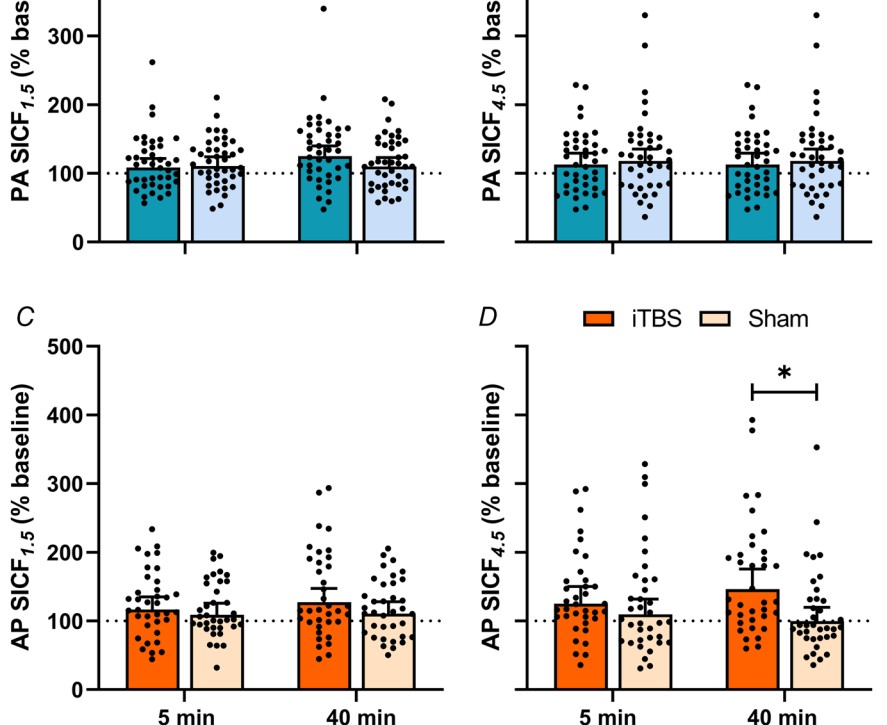

**Figure 4. Results for SICF**
*A–D*, changes in PA (blue) SICF$_{1.5}$ (*A*) and SICF$_{4.5}$ (*B*), and AP (orange) SICF$_{1.5}$ (*C*) and SICF$_{4.5}$ (*D*) following PMd iTBS (darker hue) and sham (lighter hue) at 5 and 40 min in all participants. Data show EMM [95% CI] with individual subject means. *$P < 0.05$. #$P < 0.05$ compared to 5 min. [Colour figure can be viewed at wileyonlinelibrary.com]

PMd iTBS ($\rho = -0.5$, $P = 0.001$). In particular, longer mean baseline AP-LM latencies were related to greater reductions in mean AP-LM latencies following PMd iTBS. Baseline AP-LM latencies were not related to changes in PA-LM latencies ($\rho = -0.2$, $P = 0.189$) or other changes in corticospinal or intracortical excitability (all $P > 0.05$). Similarly, age was not related to changes in corticospinal excitability or intracortical function (all $P > 0.05$).

## Discussion

In the present study, we investigated the influence of PMd on I-wave excitability in young and older adults. This was achieved by assessing changes in M1 activity following application of iTBS to PMd in young and older adults. We measured changes in corticospinal excitability ($PA_{1mV}$, $AP_{1mV}$, $PA_{0.5mV}$, $AP_{0.5mV}$), intracortical excitability (PA $SICF_{1.5}$, PA $SICF_{4.5}$, AP $SICF_{1.5}$, AP $SICF_{4.5}$) and I-wave recruitment (PA-LM latency, AP-LM latency). We found that PMd iTBS potentiated both PA and AP circuits in M1, with a stronger effect on AP circuits. Importantly, the effects of PMd iTBS on the PA circuits were less in older adults.

### PMd influence on corticospinal excitability in young and older adults

Previous work has demonstrated that application of iTBS to PMd potentiates $PA_{1mV}$ measures of M1 corticospinal excitability by ~50% in young adults, which is thought to arise from the induction of long-term potentiation (LTP)-like effects within PMd, resulting in increased M1 excitability (Meng et al., 2020). Although rTMS to PMd has previously been shown to specifically modulate PA

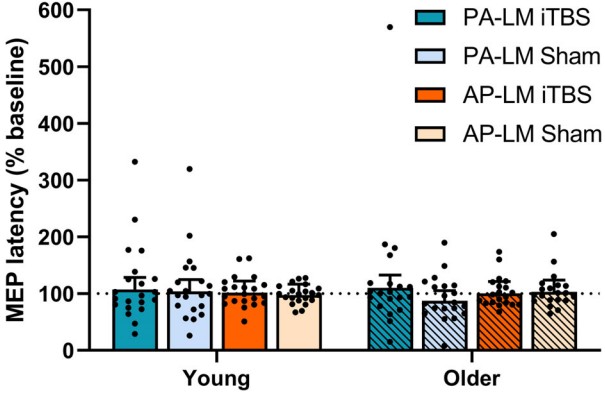

**Figure 5. Results for MEP latency**
Changes in MEP latency for PA TMS (blue) and AP TMS (orange) in young (no stripes) and older (stripes) adults following PMd iTBS (darker hue) and sham (lighter hue). Data show EMM [95% CI] with individual subject means. [Colour figure can be viewed at wileyonlinelibrary.com]

circuits (Suppa et al., 2008), our finding of a facilitation for both $PA_{1mV}$ and $AP_{1mV}$ MEP amplitudes (~20–50% increase) indicates that the LTP-like effects of iTBS on PMd extend to AP-sensitive measures of corticospinal excitability. Importantly, this facilitation is unlikely to be the result of direct activation of M1 from the spread of stimulation during iTBS over PMd, as this possibility has been tested in a previous study, which estimated the TMS intensity that reaches M1 when applying PMd TBS (Huang et al., 2009). It was reported that when this intensity was applied directly over M1, it did not influence M1 excitability (Huang et al., 2009). Given that the present study identified the location of PMd similarly to previous studies (Huang et al., 2009, 2018; Meng et al., 2020), it is therefore unlikely that M1 was directly activated during PMd iTBS.

Although facilitation of $AP_{1mV}$ MEPs after PMd iTBS was present for both groups, this effect was immediate for young adults but only observed after 40 min in older adults. Delays in neuroplastic response to NIBS interventions that directly targeted M1 have been documented previously in older adults (Fujiyama et al., 2014; Ghasemian-Shirvan et al., 2020; Opie et al., 2018), but it is unclear what mechanisms are responsible. At the synaptic level, plasticity is influenced by the interplay of excitatory (glutamate) and inhibitory (GABA) communication (Zhao et al., 2017; Ziemann et al., 2001). Furthermore, application of PMd iTBS has previously been shown to immediately reduce activity within M1 inhibitory circuits [assessed via the paired-pulse TMS measure short intracortical inhibition (SICI)], followed by a relatively delayed increase in M1 excitatory circuits [assessed via the paired-pulse TMS measure intracortical facilitation (ICF)], suggesting that the effects of PMd modulation on M1 excitability are initially driven by changes in inhibition (Meng et al., 2020). Given that a previous paired-coil TMS study has identified weakened PMd–M1 effective connectivity in older adults (Ni et al., 2015), we could therefore speculate that the delayed facilitation of $AP_{1mV}$ indirectly stemmed from age-related reductions in the sensitivity of intracortical inhibitory circuits to projections from PMd, but this will need to be clarified in future research.

The conventional interpretation of using different TMS coil orientations suggests preferential recruitment of early ($I_1$) waves with PA TMS and late ($I_3$) waves with AP TMS (Hamada et al., 2013), with either current direction able to recruit both early and late I-waves as the stimulation intensity is increased (Di Lazzaro et al., 2001, 2003). We therefore attempted to investigate the response of different I-wave circuits to PMd iTBS by recording single-pulse TMS at relatively lower stimulation intensities compared to $MEP_{1mV}$ ($PA_{0.5mV}$ and $AP_{0.5mV}$ MEPs), where PA TMS is more selective to early circuits, whereas AP TMS is more selective to late circuits (Opie et al.,

2022). Within the current study, the potentiation of both $PA_{0.5mV}$ and $AP_{0.5mV}$ MEPs (∼50–90% increase) in young adults following PMd iTBS therefore probably reflects the facilitation of early and late circuits respectively.

Although $AP_{0.5mV}$ MEPs increased following PMd iTBS in both age groups, potentiation of $PA_{0.5mV}$ was only apparent in young adults, suggesting that the direct modulatory capacity of PMd on early I-waves is specifically weakened in older adults. However, it is unclear why this effect was not shown for $PA_{1mV}$, which is also expected to recruit early circuits (Di Lazzaro et al., 1998). One possibility is that there were late circuits recruited by $PA_{1mV}$ TMS that were also facilitated by PMd iTBS, and this compensated for the weakened early circuit connections in older adults. In a similar vein, the absence of a time course delay in $AP_{0.5mV}$ facilitation in older adults could also suggest that this indirect age-related difference was specific to the early circuits recruited by $AP_{1mV}$ TMS. This suggests that the PMd influence on early I-waves may potentially be directly (reduced facilitation) and indirectly (time course delay in facilitation) affected by age. However, as these considerations are speculative, further investigations that can isolate the individual components of the I-wave circuitry, such as modifying the TMS pulse width (Hannah & Rothwell, 2017), will be useful for identifying exactly how PMd influences early and late I-wave circuits. Despite this, the current study provides new evidence that the influence of PMd on early I-waves is specifically reduced in older adults.

### PMd influence on intracortical excitability in young and older adults

In the present study, we used PA and AP SICF to investigate changes in the activity of different I-wave circuits. While SICF responses to PMd iTBS have not been previously tested, modulation of intracortical circuits indexed using paired-pulse stimulation following PMd iTBS has been reported (Meng et al., 2020). Specifically, SICI was reduced while ICF was potentiated following PMd iTBS, indicating an increase in excitability within M1 (Meng et al., 2020). In addition, given that recent literature supports preferential effects of PMd on the late I-wave circuits (Aberra et al., 2020; Volz et al., 2015), we expected that facilitation of SICF would be specific to the late I-wave measures. We found that both PA $SICF_{1.5}$ and AP $SICF_{4.5}$ were potentiated (∼20–40% increase) across both young and older adults. As the 1.5 ms ISI assesses early I-waves and 4.5 ms ISI assesses late I-waves (Opie et al., 2018; Ziemann et al., 1998), one interpretation of these results could be that PA $SICF_{1.5}$ preferentially activates early I-waves, whereas AP $SICF_{4.5}$ preferentially activates late I-waves. This finding suggests that TMS current direction and ISI contribute to improving the selectivity of SICF on different I-wave circuits. This would be consistent with our single-pulse findings for $PA_{0.5mV}$ and $AP_{0.5mV}$, and suggests that PMd iTBS may modulate both early and late I-waves.

While both PA $SICF_{1.5}$ and AP $SICF_{4.5}$ responses were potentiated following PMd iTBS, this effect occurred at 40 min and did not differ between age groups, which is inconsistent with the single-pulse findings. Despite this, it is possible that the measure of SICF was complicated by the $MEP_{1mV}$ test stimulus, which probably resulted in mixed recruitment of other I-wave circuits (Opie et al., 2021). For example, the recruitment of the late $I_2$-wave by the test stimulus may have reduced the selectivity of SICF, as the $I_2$-wave has recently been shown to be specifically influenced by ventral premotor cortex (PMv) (Casarotto et al., 2023). Furthermore, there is also a growing body of evidence to suggest that PA and AP TMS recruit distinct early and late circuits (i.e. PA- and AP-sensitive early and late I-waves) (Opie & Semmler, 2021; Spampinato et al., 2020). While speculative, it is possible that PA $SICF_{1.5}$ and AP $SICF_{4.5}$ may have activated other I-waves that are not modulated by PMd iTBS, limiting the extent of facilitation. This may also explain why PA $SICF_{4.5}$ and AP $SICF_{1.5}$ responses were not facilitated, as these two measures may have recruited a greater proportion of these I-waves. Finally, the large variability of SICF responses (shown at baseline) may have also limited findings following PMd iTBS. Specifically, the peak facilitation of the late $I_3$-wave is known to be highly variable between age groups, with the optimal peak of the $I_3$-wave shown to be ∼4.1–4.3 ms in young adults, whereas the optimal peak in older adults is ∼5 ms (Opie et al., 2018). The use of 4.5 ms ISI for both age groups in the present study was a deliberate decision to allow for comparisons of the late I-waves between age groups (as shown by similar $SICF_{4.5}$ responses at baseline). However, this may have also contributed to the absence of differences between age groups following PMd iTBS, as optimal ISIs for both groups were not used. Consequently, while our SICF findings suggest that PMd influences both early and late I-wave circuits, it will be necessary to identify TMS measures that are comparable between age groups and, possibly, more sensitive to age-related effects (Semmler et al., 2021).

### PMd influence on I-wave recruitment in young and older adults

Previous studies have investigated the ability to recruit early (PA) and late (AP) I-waves with TMS by comparing PA and AP TMS latencies relative to the latencies of direct corticospinal activation (PA-LM, early; AP-LM, late). This measure demonstrates that PA-LM latencies (∼1.5 ms) are shorter than AP-LM latencies (∼3 ms), providing an index of early and late I-wave recruitment (Hamada et al., 2013).

While the baseline MEP latencies of young adults in the present study were similar to those reported previously (Hamada et al., 2013; Opie et al., 2018; Volz et al., 2015), we found that baseline PA-LM latencies were longer in older adults compared to young adults, inconsistent with previous work that reported no age-related differences (Opie et al., 2018). Longer PA-LM latencies (>2 ms) have been suggested to reflect increased activation of the PA-sensitive late I-wave circuits (Hamada et al., 2013). While speculative, this may explain why PMd iTBS did not modulate $PA_{0.5mV}$ in older adults, as there was greater activation of the PA-sensitive late circuits less affected by PMd iTBS. However, as our correlation analyses did not indicate any relationships between the baseline PA-LM latency and changes in early I-wave excitability, further studies investigating age-related changes in the recruitment of early I-waves will be necessary.

Importantly, previous work has suggested that the ability to recruit late (AP) I-waves with TMS is associated with the strength of PMd–M1 connectivity (Volz et al., 2015). Furthermore, it has been previously demonstrated that AP-LM latency can be specifically modulated by M1 iTBS, which was thought to influence the AP inputs that originate from PMd (Volz et al., 2019). We therefore also examined the changes in PA-LM and AP-LM latencies after PMd iTBS. While no changes in the latency measures were observed, longer baseline AP-LM latencies were related to stronger reductions in AP-LM latency following PMd iTBS, which complements the specific modulation of AP-LM latency following M1 iTBS reported previously (Volz et al., 2019). Although this could be considered evidence that PMd may more strongly influence late (AP) circuits (Aberra et al., 2020), we found no evidence of changes in AP-LM latency or other relationships between baseline AP-LM latency and changes in M1 excitability. Alternatively, we could speculate that the effects of PMd iTBS on MEP latency are limited to immediately after the intervention, as the MEP latency measures were recorded at the beginning and end of the experimental session (at least 1 h either side of PMd iTBS). However, this was done to avoid complications involving the effects of muscle activation on neuroplasticity responses (Goldsworthy et al., 2015; Huang et al., 2008; Thirugnanasambandam et al., 2011). Therefore, while this finding may suggest a selective influence of PMd on late circuits (Aberra et al., 2020), further studies are required to characterise the relationship between I-wave recruitment and neuroplastic changes involving PMd–M1 communication.

In conclusion, the application of iTBS over PMd potentiated corticospinal and intracortical excitability. Importantly, our results provide new evidence that PMd targets both early and late I-wave circuits, with some evidence showing that this communication is stronger for the late I-waves. Critically, we also provide new evidence that the effect of PMd on M1 excitability is specifically reduced for the early circuits in older adults. It will therefore be useful in future studies to investigate how this age-related difference in the PMd modulation of early and late I-waves influences M1 plasticity and motor skill learning in young and older adults.

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

## Additional information

### Data availability statement

Data from this study will be made available to qualified investigators upon reasonable request to the corresponding author.

### Competing interests

The authors have no conflicts of interest to declare.

### Author contributions

W.-Y.L. Conception or design of the work; Acquisition, analysis or interpretation of data for the work; Drafting the work or revising it critically for important intellectual content; Final approval of the version to be published; Agreement to be

accountable for all aspects of the work. G.O. Conception or design of the work; Acquisition, analysis or interpretation of data for the work; Drafting the work or revising it critically for important intellectual content; Final approval of the version to be published; Agreement to be accountable for all aspects of the work. U.Z. Conception or design of the work; Acquisition, analysis or interpretation of data for the work; Drafting the work or revising it critically for important intellectual content; Final approval of the version to be published; Agreement to be accountable for all aspects of the work. J.S. Conception or design of the work; Acquisition, analysis or interpretation of data for the work; Drafting the work or revising it critically for important intellectual content; Final approval of the version to be published; Agreement to be accountable for all aspects of the work.

## Funding

Australian Research Council Discovery Project: Wei-Yeh Liao, George M. Opie, Ulf Ziemann, John G. Semmler, DP200101009; DHAC | National Health and Medical Research Council (NHMRC): George M. Opie, APP1139723.

## Acknowledgements

Support was provided by an Australian Research Council Discovery Projects Grant (grant number DP200101009). G.M.O. was supported by a National Health and Medical Research Council early career fellowship (APP1139723).

Open access publishing facilitated by The University of Adelaide, as part of the Wiley - The University of Adelaide agreement via the Council of Australian University Librarians.

## Keywords

ageing, dorsal premotor cortex, transcranial magnetic stimulation

## Supporting information

Additional supporting information can be found online in the Supporting Information section at the end of the HTML view of the article. Supporting information files available:

**Statistical Summary Document**
**Peer Review History**

