## [Peer Review History · The Journal of Physiology]

Modulation of dorsal premotor cortex differentially influences I-wave excitability in primary motor cortex of young and older adults

Wei-Yeh Liao, George M. Opie, Ulf Ziemann, and John G. Semmler

DOI: 10.1113/JP284204

Corresponding author(s): Wei-Yeh Liao (wei-yeh.liao@adelaide.edu.au)

The following individual(s) involved in review of this submission have agreed to reveal their identity: Roisin McMackin (Referee #1); Matt J.N. Brown (Referee #2)

Review Timeline:

Submission Date:	01-Dec-2022
Editorial Decision:	26-Jan-2023
Revision Received:	22-Feb-2023
Editorial Decision:	12-Apr-2023
Revision Received:	13-Apr-2023
Accepted:	12-May-2023

Senior Editor: Richard Carson

Reviewing Editor: Charlotte Stagg

Transaction Report:

Dear Mr Liao,

Re: JP-RP-2022-284204 "Modulation of dorsal premotor cortex differentially influences I-wave excitability in primary motor cortex of young and older adults" by Wei-Yeh Liao, George M. Opie, Ulf Ziemann, and John G. Semmler

Thank you for submitting your manuscript to The Journal of Physiology. It has been assessed by a Reviewing Editor and by 2 expert referees and we are pleased to tell you that it is acceptable for publication following satisfactory revision.

REVISION CHECKLIST:

- 'Potential Cover Art' for consideration as the issue's cover image

- Appropriate Supporting Information (Video, audio or data set: see https://jp.msubmit.net/cgi-bin/main.plex?form_type=display_requirements#supp).

We look forward to receiving your revised submission.

Yours sincerely,

Richard Carson
Senior Editor
The Journal of Physiology

REQUIRED ITEMS

-Author photo and profile. First (or joint first) authors are asked to provide a short biography (no more than 100 words for one author or 150 words in total for joint first authors) and a portrait photograph. These should be uploaded and clearly labelled with the revised version of the manuscript. See Information for Authors for further details.

â€"The Journal of Physiology funds authors of provisionally accepted papers to use the premium BioRender site to create high resolution schematic figures. Follow this link and enter your details and the manuscript number to create and download figures. Upload these as the figure files for your revised submission. If you choose not to take up this offer we require figures to be of similar quality and resolution. If you are opting out of this service to authors, state this in the Comments section on the Detailed Information page of the submission form. The link provided should only be used for the purposes of this submission. Authors will be charged for figures created on this premium BioRender account if they are not related to this manuscript submission.

-Please upload separate high-quality figure files via the submission form.

-We invite you to include a Translational Perspective paragraph in your manuscript. This should be included in the main body of the manuscript after the Acknowledgements. It should describe the wider translational implications of the work, in plain English, for a broad scientific audience. Please use the following guidelines to prepare a Translational perspective of your paper https://jp.msubmit.net/cgi-bin/main.plex?form_type=display_requirements#authortranspersp The Translational perspective should not exceed 250 words in total and should be presented as a single paragraph. Abbreviations and technical terms must be defined as briefly and simply as possible the first time they are used, unless they are generally/easily understood, e.g. ECG, HIV/AIDS, K+ channel. Use language that can be understood by scientists or clinicians with a general knowledge of the topic addressed. Ensure the paragraph includes the hypothesis tested in the paper and accurately reflects the findings of the paper and the implications for future research. State the word count of the Translational perspective paragraph.

-A Statistical Summary Document, summarising the statistics presented in the manuscript, is required upon revision. It must be on the Journal's template, which can be downloaded from the link in the Statistical Summary Document section here: https://jp.msubmit.net/cgi-bin/main.plex?form_type=display_requirements#statistics

-Please include an Abstract Figure file, as well as the figure legend text within the main article file. The Abstract Figure is a piece of artwork designed to give readers an immediate understanding of the research and should summarise the main conclusions. If possible, the image should be easily 'readable' from left to right or top to bottom. It should show the physiological relevance of the manuscript so readers can assess the importance and content of its findings. Abstract Figures should not merely recapitulate other figures in the manuscript. Please try to keep the diagram as simple as possible and without superfluous information that may distract from the main conclusion(s). Abstract Figures must be provided by authors no later than the revised manuscript stage and should be uploaded as a separate file during online submission labelled as File Type 'Abstract Figure'. Please ensure that you include the figure legend in the main article file. All Abstract Figures should be created using BioRender. Authors should use The Journal's premium BioRender account to export high-resolution images. Details on how to use and access the premium account are included as part of this email.

-Please include a full title page as part of your article (Word) file (containing title, authors, affiliations, corresponding author name and contact details, keywords, and running title).

EDITOR COMMENTS

Reviewing Editor:

Thank you for submitting your manuscript to the Journal of Physiology where it has been carefully reviewed by 2 expert reviewers and by the editorial team. As you can see, while all appreciate the question you ask and are broadly positive about the paper and its potential impact on the field, both reviewers have a number of comments that will need to be addressed.

In addition, the editorial team make the following points:

The key points and title are appropriate. The ToC category is correct. The reviewers raise a few points that would benefit from some clarification, but broadly the introduction and methods are clear and well written. The figures show individual data points and n is clear - thank you. The authors use a 95% Confidence Intervals rather than Standard Deviation throughout but this is clearly stated and appropriate. The figures and tables are clear. The manuscript is well written.

Please ensure that precise p values are stated throughout, rather than just > or <.

REFEREE COMMENTS

Referee #1:

This is a very nice paper, nice work and clearly explained. I have a few queries for the authors:

What, more specifically, does "low-level activation of the right FDI" refer to? Rejection of trials with baseline EMG activity above 25uV is mentioned later on, so was AMT recorded during contraction with maximum amplitudes of of 25uV?

"Following AMT, the stimulus intensities producing a standard MEP amplitude approximating 1 mV (MEP1mV; PA1mV and AP1mV), in addition to an MEP amplitude approximating 0.5 mV (MEP0.5mV; PA0.5mV and AP0.5mV), when averaged over 20 trials, were identified" - What was the procedure for this? For example, was the stimulator intensity systematically varied after averaging 20 trials at each intensity? Was PEST used?

"For all measures of SICF, individual MEP amplitudes produced by paired-pulse stimulation were expressed as a percentage of the mean MEP amplitude produced by single-pulse stimulation recorded at baseline (Cash et al., 2009; Liao et al., 2022), as Spearman's correlation analysis revealed that post-intervention increases in single-pulse test MEP amplitudes predicted increases in the paired-pulse MEP amplitudes ($\rho = 0.8$, $P < 0.05$)" - I'm confused by this approach. Say for example that iTBS increases single pulse MEP amplitude and increases paired-pulse MEP amplitude, should taking the conditioned vs unconditioned average MEP amplitude ratio not correct for this? By comparing pre-iTBS single pulse with post-iTBS paired pulse are you not then incorporating the effect of iTBS on single pulse MEPs into your SICF measure?

"Correlations are presented as Spearman's ρ with significance set at $P < 0.00167$ following Bonferroni correction." - Why use Bonferroni? At a correction across 30 comparisons to achieve a corrected P threshold of 0.00167 you are likely overcorrecting. What about adaptive FDR?

The use of the term "low-intensity" is confusing/misleading, I presume this refers to the use of peri-threshold intensities, but it

could be interpreted as sub-threshold or as low intensity absolute values (for example <30% MSO)

What did participants do in the 1 hour intervals before and after iTBS and in the gap between 5 40 min post-iTBS measurements?

"We found that both PA SICF1.5 and AP SICF4.5 were potentiated (~20-40% increase) across both young and older adults. As the 1.5ms ISI assesses early I-waves and 4.5 ms ISI assesses late I-waves (Ziemann et al., 1998;Opie et al., 2018), one interpretation of these results could be that PA SICF1.5 activates PA early I-waves, whereas AP SICF4.5 activates AP late I-waves. This would be consistent with our single-pulse findings for PA0.5mV and AP0.5mV, and suggests that PMd iTBS may

preferentially modulate both PA early and AP late I-waves." - Two points on this. First, I think the authors should remove the reference to "early PA" and late AP" throughout, as it implies that PA only activates early I waves, and AP only activates late I waves, in addition to suggesting that the late I waves activated by one orientation are different from the late I waves activated by another (ditto for early I waves). It would be better to simply refer to early or late I waves when naming the I waves. Second, it is later mentioned that "it is possible that the measure of SICF was complicated by the MEP1mV test stimulus, which likely resulted in mixed recruitment of I wave circuits (Opie et al., 2021)", which is a good point. Could the detection of an effect on SICF4.5ms using AP and on SICF1.5ms using PA reflect that these different orientations preferentially (but not solely) activate the I waves that are intended to be interrogated by these different ISIs, and therefore there is more to be facilitated/inhibited. For example, if AP orientation engages late I wave circuits more so than AP orientation, then it seems logical that AP orientation would be a better orientation to use when attempting to facilitate the late I waves with SICF4.5ms, and would be more sensitive to effects of iTBS on said measure.

Overall I think there needs to be a bit of caution regarding intermixing reference to early vs late I waves and AP vs PA orientation, as at suprathreshold intensities such as those evoking 1mV MEPs, both I waves will be engaged by both orientations. More emphasis on "preferential" engagement is warranted.

Referee #2:

Comments on Liao et al.

Liao et al. investigated the effects of PMd iTBS on MEP amplitudes and latencies with both PA and AP currents as well as 0.5mV and 1mV TS. The investigators also examined PA and AP effect for SICF at 1.5 and 4.5 ms ISIs. Overall, younger adults had higher MEP amplitudes, shorter MEP latencies and higher stimulation intensities for PA at 0.5 and 1mV. The main findings related to iTBS were significant increases iTBS compared to sham, with the only prominent group difference that older adults failed to demonstrate a significance increase compared to sham at 5 minutes post (but younger adults did).

Overall, the manuscript was very well written, easy to follow and examined an interesting research question that was supported by a hypothesis. The majority of my comments are relatively minor and aimed at improving the reader's understanding and readability of the manuscript.

Comments:

Although Figure 1 alludes to it, it is not clearly stated in text that the left PMd was always targeting. Please make this clear in the methods section for iTBS

In the data analysis section you state "as Spearman's correlation analysis revealed

that post-intervention increases in single-pulse test MEP amplitudes predicted increases in the

paired-pulse MEP amplitudes ($\rho = 0.8$, $P < 0.05$ ". Are these results from your work or the previous work? If your work, please include them in the results section.

A visual from BrainSight for the stimulation location of PMd for all participants on standard MNI-ICBM152 template as well as a table with MNI coordinates (this could be supplemental figures and table). I think this is particularly important since PMd location was not based off function or structure from individualized MRIs.

In the data analyses, you collapsed the MEP latency data across multiple stimulation intensities. Please provide data (even in supplemental data) that there were no differences in MEP latency across PA1mV, AP 1mVn PA0.5mV and AP0.5mV.

Why did the analyses investigating the changes in corticospinal excitability following the intervention using baseline-normalized values not look at the PA1mV, AP 1mVn PA0.5mV and AP0.5mV altogether? This is the statistical method you used for the SICF measures.

In the analyses, you stated you used Spearman's rank-order correlation, but then stated you regressed variables. Did you correlate two variables (which is what Spearman's rank-order correlation does) or regress? If you did a regression, please state proper analyses.

Due to the inability to record several of the measures, it would be important for the authors to include which participants (example participant code) and their demographic variables were excluded, and the remaining demographic variables for those left in the analyses (again, this is something that could be put in supplemental data). I believe it is interesting to the readers to understand what may have been different about these participants and why might that have contributed to not being able to record these measures.

I am curious why you also didn't include PA1mV over time in young vs. old within Figure 2? I know there was not statistically significant difference, but it provides a nice visual contrast to the AP1mV already included in Figure 2 (part C)

Older adults failed to demonstrate they were different with iTBS compared to sham with PA0.5mV but did demonstrate for AP0.5mV. When taken into consideration with the older adults only demonstrating a significant increase in AP1mV 40 (vs. 5) minutes post, what can that all mean? Are these changes correlated?

No differences between young vs. older adults for SICF. PA1.5mV increased at 40 minutes and compared to sham while AP4.5mV increased after iTBS compared to sham. APSICF1.5 did not change but there was no discussion of PA4.5mV results? Please include a statement regarding PA4.5mV SICF.

In regards to Figure 5b and significant correlation analyses - there is no significant difference in MEP latency as stated in the above paragraph. Why is this correlation relevant as there is no significant change? I think this might just confuse the reader and it doesn't really add to the story so I think you should delete it.

In the last sentence of your discussion you write "Importantly, the effects of PMd iTBS on the I-wave circuits were less in older compared to young adults." Although it is true your results support difference in older compared to younger adults, I don't believe a single analysis post-iTBS revealed a direct difference between young vs. old, so this is misleading.

END OF COMMENTS

Confidential Review

01-Dec-2022

Response to Reviewing Editor:

1. *Please ensure that precise p values are stated throughout, rather than just > or <.*

Precise *P*-values (to three significant figures) are stated in the manuscript, except where multiple comparisons are covered by a single statement or *P*-values are less than 0.001.

Response to Referee #1:

1. *What, more specifically, does "low-level activation of the right FDI" refer to? Rejection of trials with baseline EMG activity above 25 μ V is mentioned later on, so was AMT recorded during contraction with maximum amplitudes of 25 μ V?*

Low-level activation refers to ~10% voluntary contraction of the right FDI. Participants were instructed to maintain ~10% contraction of the muscle during active TMS recordings (Hamada *et al.*, 2013; D'Ostilio *et al.*, 2016). This has now been stated in the manuscript (page 7). In addition, rejection of trials due to EMG activity above 25 μ V was only completed for trials recorded in the resting muscle. This has now been clarified (page 9).

2. *"Following AMT, the stimulus intensities producing a standard MEP amplitude approximating 1 mV (MEP_{1mV} ; PA_{1mV} and AP_{1mV}), in addition to an MEP amplitude approximating 0.5 mV ($MEP_{0.5mV}$; $PA_{0.5mV}$ and $AP_{0.5mV}$), when averaged over 20 trials, were identified" - What was the procedure for this? For example, was the stimulator intensity systematically varied after averaging 20 trials at each intensity? Was PEST used?*

During these measures, we estimated an intensity and sampled 20 trials for each of the conditions. The intensity was adjusted and the trials were re-recorded if the average MEP amplitude did not fall in the range of 0.8-1.2 mV for MEP_{1mV} and 0.4-0.6 mV for $MEP_{0.5mV}$. This process has now been stated in the manuscript (pages 7-8).

3. *"For all measures of SICF, individual MEP amplitudes produced by paired-pulse stimulation were expressed as a percentage of the mean MEP amplitude produced by single-pulse stimulation recorded at baseline (Cash *et al.*, 2009; Liao *et al.*, 2022), as Spearman's correlation analysis revealed that post-intervention increases in single-pulse test MEP amplitudes predicted increases in the paired-pulse MEP amplitudes ($\rho = 0.8$, $P < 0.05$)" - I'm confused by this approach. Say for example that iTBS increases single pulse MEP amplitude and increases paired-pulse MEP amplitude, should taking the conditioned vs unconditioned average MEP amplitude ratio not correct for this? By comparing pre-iTBS single pulse with post-iTBS paired pulse are you not then incorporating the effect of iTBS on single pulse MEPs into your SICF measure?*

It has previously been shown that there is a correlation between the increase in single-pulse MEP amplitude and increase in SICF after I-wave periodicity TMS (iTMS), and that normalising to the post-iTMS MEP amplitude underestimates the change in excitability of the I-wave generating networks (Cash *et al.*, 2009). To illustrate this point theoretically, if there is a two-fold increase in single-pulse MEP amplitude after an intervention (e.g. 1 to 2 mV), a two-fold increase in paired-pulse TMS (e.g. 2 to 4 mV) would result in no change in SICF after the intervention (when normalising to the post-intervention test MEP), even though

there has been an absolute increase in SICF. Consequently, we also found a significant relationship between single- and paired-pulse MEP amplitude post-iTBS. Therefore, to avoid underestimating the effect of iTBS on I-wave facilitation, we have expressed post-iTBS SICF responses as a percentage of pre-iTBS single-pulse MEP amplitude, which is in line with previous studies (Cash *et al.*, 2009; Opie *et al.*, 2021; Liao *et al.*, 2022). We have now explained this rationale and process in the manuscript (pages 9-10).

4. *“Correlations are presented as Spearman’s ρ with significance set at $P < 0.00167$ following Bonferroni correction.” – Why use Bonferroni? At a correction across 30 comparisons to achieve a corrected P threshold of 0.00167 you are likely overcorrecting. What about adaptive FDR?*

Thank you to the reviewer for this suggestion. We have changed the correlation analysis to include FDR with Benjamini-Hochberg procedure. The details are now included in the methods and results section of the manuscript (pages 11, 15).

5. *The use of the term “low-intensity” is confusing/misleading, I presume this refers to the use of peri-threshold intensities, but it could be interpreted as sub-threshold or as low intensity absolute values (for example $<30\%$ MSO).*

We have removed the term “low-intensity” throughout the manuscript and directly referenced the measure as $MEP_{0.5mV}$.

6. *What did participants do in the 1-hour intervals before and after iTBS and in the gap between 5 and 40 min post-iTBS measurements?*

Participants remained seated and relaxed during breaks between TMS blocks as per instructions, but were allowed to read or use mobile devices using their left hand. Participants were specifically asked to maintain relaxation of the right hand during the experiment and this was monitored throughout. This is now stated in the manuscript (page 7).

7. *“We found that both PA SICF_{1.5} and AP SICF_{4.5} were potentiated (~20-40% increase) across both young and older adults. As the 1.5ms ISI assesses early I-waves and 4.5 ms ISI assesses late I-waves (Ziemann *et al.*, 1998; Opie *et al.*, 2018), one interpretation of these results could be that PA SICF_{1.5} activates PA early I-waves, whereas AP SICF_{4.5} activates AP late I-waves. This would be consistent with our single-pulse findings for PA_{0.5mV} and AP_{0.5mV}, and suggests that PMd iTBS may preferentially modulate both PA early and AP late I-waves.” - Two points on this. First, I think the authors should remove the reference to “early PA” and late AP” throughout, as it implies that PA only activates early I waves, and AP only activates late I waves, in addition to suggesting that the late I waves activated by one orientation are different from the late I waves activated by another (ditto for early I waves). It would be better to simply refer to early or late I waves when naming the I waves. Second, it is later mentioned that “it is possible that the measure of SICF was complicated by the MEP_{1mV} test stimulus, which likely resulted in mixed recruitment of I wave circuits (Opie *et al.*, 2021)”, which is a good point. Could the detection of an effect on SICF_{4.5} using AP and on SICF_{1.5} using PA reflect that these different orientations preferentially (but not solely) activate the I-waves that are intended to be interrogated by these different ISIs, and therefore there is more to be facilitated/inhibited. For example, if AP orientation engages late I wave circuits more*

so than PA orientation, then it seems logical that AP orientation would be a better orientation to use when attempting to facilitate the late I waves with SICF_{4,5}, and would be more sensitive to effects of iTBS on said measure.

We agree with the reviewer's interpretation of our findings. We have therefore removed the explanation of early and late PA, and early and late AP I-waves in favour of early and late I-waves, in addition to altering the discussion of SICF (pages 16-19). However, we felt that it was important to highlight the growing literature showing the possibility that PA and AP TMS can activate distinct early and late I-waves, and have briefly covered this point in the discussion of SICF (pages 17-18).

Response to Referee #2:

1. *Although Figure 1 alludes to it, it is not clearly stated in text that the left PMd was always targeting. Please make this clear in the methods section for iTBS*

Targeting of left PMd has now been stated in the methods section for iTBS (pages 8-9).

2. *In the data analysis section you state "as Spearman's correlation analysis revealed that post-intervention increases in single-pulse test MEP amplitudes predicted increases in the paired-pulse MEP amplitudes ($\rho = 0.8, P < 0.05$)". Are these results from your work or the previous work? If your work, please include them in the results section.*

The result of this correlation analysis is now included in the results (page 14).

3. *A visual from BrainSight for the stimulation location of PMd for all participants on standard MNI-ICBM152 template as well as a table with MNI coordinates (this could be supplemental figures and table). I think this is particularly important since PMd location was not based off function or structure from individualized MRIs.*

Our PMd location was measured relative to M1 hotspot, and we recorded these locations in BrainSight in order to maintain consistent coil positioning when delivering PMd iTBS. We have clarified this in the methods section of the manuscript (page 9), in addition to including a visual and table of MNI coordinates of both M1 and PMd for all participants (See supplementary materials).

4. *In the data analyses, you collapsed the MEP latency data across multiple stimulation intensities. Please provide data (even in supplemental data) that there were no differences in MEP latency across PA1mV, AP 1mVn PA0.5mV and AP0.5mV.*

We did not assess MEP latency from the single-pulse TMS data used for MEP amplitude analysis. Rather, our method for calculating MEP latency data follows the accepted practice of using separate recordings of 110% AMT_{PA} for PA latency, 110% AMT_{AP} for AP latency, and 150% of AMT_{LM} for LM latency (Hamada *et al.*, 2013). Together, the average latencies from these active muscle recordings were used to calculate PA-LM and AP-LM latencies (Hamada *et al.*, 2013). We have clarified this process in the methods section for data analysis of the manuscript (page 8-9).

5. *Why did the analyses investigating the changes in corticospinal excitability following the intervention using baseline-normalized values not look at the PA_{1mV}, AP_{1mV}.*

PA_{0.5mV} and AP_{0.5mV} altogether? This is the statistical method you used for the SICF measures.

Our statistical method for both single-pulse and paired-pulse measures post-intervention were the same. Specifically, each individual measure (PA_{1mV}, AP_{1mV}, PA_{0.5mV}, AP_{0.5mV}, PA SICF_{1.5}, PA SICF_{4.5}, AP SICF_{1.5}, and AP SICF_{4.5}) was examined using separate models (eight total) that investigated the factors of session (iTBS, sham), time (5 minutes, 40 minutes), and age (young, older). We have clarified this process in the methods section for statistical analysis of the manuscript (pages 10-11).

- 6. In the analyses, you stated you used Spearman's rank-order correlation, but then stated you regressed variables. Did you correlate two variables (which is what Spearman's rank-order correlation does) or regress? If you did a regression, please state proper analyses.*

We have corrected this section of statistical analysis to reflect Spearman's rank-order correlation analysis (page 11).

- 7. Due to the inability to record several of the measures, it would be important for the authors to include which participants (example participant code) and their demographic variables were excluded, and the remaining demographic variables for those left in the analyses (again, this is something that could be put in supplemental data). I believe it is interesting to the readers to understand what may have been different about these participants and why might that have contributed to not being able to record these measures.*

We agree with the reviewer that there are demographic variables that may contribute to the difficulty of recording certain TMS measures. However, the demographic variables recorded in the present study (age, sex, and activation threshold) do not appear to show any meaningful contribution. We have therefore chosen to briefly highlight these details for the participants in whom we were unable to make TMS recordings in the results section of the manuscript (page 11).

- 8. I am curious why you also didn't include PA_{1mV} over time in young vs. old within Figure 2? I know there was not statistically significant difference, but it provides a nice visual contrast to the AP_{1mV} already included in Figure 2 (part C).*

Thank you to the reviewer for this suggestion. We have now added the graph for PA_{1mV} over time in young and older adults (Fig. 2C).

- 9. Older adults failed to demonstrate that they were different with iTBS compared to sham with PA_{0.5mV} but did demonstrate for AP_{0.5mV}. When taken into consideration with the older adults only demonstrating a significant increase in AP_{1mV} 40 (vs. 5) minutes post, what can that all mean? Are these changes correlated?*

Thank you to the reviewer for highlighting this point. We have suggested that the absence of facilitation for PA_{0.5mV} in older adults indicates a weakened direct influence of PMd on early I-wave excitability. We also speculated that the time course delay in facilitation for AP_{1mV} in older adults may have been the result of an indirect mechanism, due to the weakened sensitivity in inhibitory circuits that received PMd projections, as previous work has shown that changes in excitatory (I-wave) circuits following PMd iTBS are preceded by changes in

inhibitory circuits (Meng *et al.*, 2020). Given that AP_{1mV} likely recruited more early I-waves compared to AP_{0.5mV}, this suggests that there may be both direct (absence of facilitation) and indirect effects (time course delay in facilitation) of ageing on PMd communication with early I-waves. We have now included this interpretation in the revised manuscript (pages 16-17).

10. No differences between young vs. older adults for SICF PA1.5mV increased at 40 minutes and compared to sham while AP SICF 4.5mV increased after iTBS compared to sham. AP SICF1.5 did not change but there was no discussion of PASICF 4.5mV results? Please include a statement regarding PA4.5mV SICF.

We have now included a discussion on both PA SICF_{4.5} and AP SICF_{1.5} (pages 17-18).

11. In regards to Figure 5b and significant correlation analyses - there is no significant difference in MEP latency as stated in the above paragraph. Why is this correlation relevant as there is no significant change? I think this might just confuse the reader and it doesn't really add to the story so I think you should delete it.

We agree with the reviewer and have removed Figure 5B and modified our discussion to better reflect this point (page 19).

12. In the last sentence of your discussion you write "Importantly, the effects of PMd iTBS on the I-wave circuits were less in older compared to young adults." Although it is true your results support difference in older compared to younger adults, I don't believe a single analysis post-iTBS revealed a direct difference between young vs. old, so this is misleading.

We have reworded this sentence to more accurately reflect our findings related to differences in iTBS effects for young and older adults (page 15).

References

- Cash RFH, Benwell NM, Murray K, Mastaglia FL & Thickbroom GW. (2009). Neuromodulation by paired-pulse TMS at an I-wave interval facilitates multiple I-waves. *Exp Brain Res* **193**, 1-7.
- D'Ostilio K, Goetz SM, Hannah R, Ciocca M, Chieffo R, Chen J-CA, Peterchev AV & Rothwell JC. (2016). Effect of coil orientation on strength–duration time constant and I-wave activation with controllable pulse parameter transcranial magnetic stimulation. *Clin Neurophysiol* **127**, 675-683.
- Hamada M, Murase N, Hasan A, Balaratnam M & Rothwell JC. (2013). The Role of Interneuron Networks in Driving Human Motor Cortical Plasticity. *Cereb Cortex* **23**, 1593-1605.
- Liao W-Y, Sasaki R, Semmler JG & Opie GM. (2022). Cerebellar transcranial direct current stimulation disrupts neuroplasticity of intracortical motor circuits. *PLoS One* **17**, e0271311.
- Meng H-J, Cao N, Zhang J & Pi Y-L. (2020). Intermittent theta burst stimulation facilitates functional connectivity from the dorsal premotor cortex to primary motor cortex. *PeerJ* **8**, e9253-e9253.
- Opie GM, Sasaki R, Hand BJ & Semmler JG. (2021). Modulation of Motor Cortex Plasticity by Repetitive Paired-Pulse TMS at Late I-Wave Intervals Is Influenced by Intracortical Excitability. *Brain Sci* **11**, 121.

Dear Mr Liao,

Re: JP-RP-2023-284204R1 "Modulation of dorsal premotor cortex differentially influences I-wave excitability in primary motor cortex of young and older adults" by Wei-Yeh Liao, George M. Opie, Ulf Ziemann, and John G. Semmler

Thank you for submitting your manuscript to The Journal of Physiology. It has been assessed by a Reviewing Editor and by 2 expert referees and we are pleased to tell you that it is acceptable for publication following satisfactory revision.

REVISION CHECKLIST:

- 'Potential Cover Art' for consideration as the issue's cover image

- Appropriate Supporting Information (Video, audio or data set: see https://jp.msubmit.net/cgi-bin/main.plex?form_type=display_requirements#supp).

We look forward to receiving your revised submission.

Yours sincerely,

Richard Carson
Senior Editor
The Journal of Physiology

REQUIRED ITEMS

- The Journal of Physiology funds authors of provisionally accepted papers to use the premium BioRender site to create high resolution schematic figures. Follow this link and enter your details and the manuscript number to create and download figures. Upload these as the figure files for your revised submission. If you choose not to take up this offer we require figures to be of similar quality and resolution. If you are opting out of this service to authors, state this in the Comments section on the Detailed Information page of the submission form. The link provided should only be used for the purposes of this submission. Authors will be charged for figures created on this premium BioRender account if they are not related to this manuscript submission.

- Your paper contains Supporting Information of a type that we no longer publish. Any information essential to an understanding of the paper must be included as part of the main manuscript and figures. The only Supporting Information that we publish are video and audio, 3D structures, program codes and large data files. Your revised paper will be returned to you if it does not adhere to our Supporting Information Guidelines.

- We invite you to include a Translational Perspective paragraph in your manuscript. This should be included in the main body of the manuscript after the Acknowledgements. It should describe the wider translational implications of the work, in plain English, for a broad scientific audience. Please use the following guidelines to prepare a Translational perspective of your paper https://jp.msubmit.net/cgi-bin/main.plex?form_type=display_requirements#authortranspersp The Translational perspective should not exceed 250 words in total and should be presented as a single paragraph. Abbreviations and technical terms must be defined as briefly and simply as possible the first time they are used, unless they are generally/easily understood, e.g. ECG, HIV/AIDS, K⁺ channel. Use language that can be understood by scientists or clinicians with a general knowledge of the topic addressed. Ensure the paragraph includes the hypothesis tested in the paper and accurately reflects the findings of the paper and the implications for future research. State the word count of the Translational perspective paragraph.

- Papers must comply with the Statistics Policy: https://jp.msubmit.net/cgi-bin/main.plex?form_type=display_requirements#statistics.

In summary:

- If $n \leq 30$, all data points must be plotted in the figure in a way that reveals their range and distribution. A bar graph with data points overlaid, a box and whisker plot or a violin plot (preferably with data points included) are acceptable formats.

- If $n > 30$, then the entire raw dataset must be made available either as supporting information, or hosted on a not-for-profit repository e.g. FigShare, with access details provided in the manuscript.

- 'n' clearly defined (e.g. x cells from y slices in z animals) in the Methods. Authors should be mindful of pseudoreplication.

- All relevant 'n' values must be clearly stated in the main text, figures and tables, and the Statistical Summary Document (required upon revision).

- The most appropriate summary statistic (e.g. mean or median and standard deviation) must be used. Standard Error of the Mean (SEM) alone is not permitted.

- Exact p values must be stated. Authors must not use 'greater than' or 'less than'. Exact p values must be stated to three significant figures even when 'no statistical significance' is claimed.

- Statistics Summary Document completed appropriately upon revision.

EDITOR COMMENTS

Reviewing Editor:

Thank you for re-submitting your work to the Journal of Physiology, which has been carefully reviewed by the original 2 expert reviewers and the editorial team. The reviewers are happy that you have adequately addressed their comments and have no remaining concerns. Although many of the statistical questions have been resolved, there are a few points that still need to be addressed.

In addition, thank you for making most of the p values exact. However, there are still some that are shown as < or >, and are not describing the results of multiple tests (e.g. In addition, baseline MEP latencies differed between coil orientations ($F_{1,156} = 247.41$, $P < 0.05$), with shorter PA-LM latencies compared to AP-LM latencies (EMD = 1.9 ms [1.6, 2.1], $P < 0.05$)).

p-values will need to be given precisely throughout before the paper can be accepted for publication. In addition, while it is good to see all the data points on the figures, the error bars are not described - I presume these are SD, but this is not stated. Please could the authors clarify for each figure.

Senior Editor:

Please ensure that the basis of the calculation of errors bars is stated in all instances (and that this is bodied in standard deviations, unless there are grounds for showing confidence intervals). Please also see the note from the Reviewing Editor concerning the reporting of p values.

REFEREE COMMENTS

Referee #1:

I am satisfied with the responses to my previous queries and the associated manuscript amendments.

Referee #2:

The authors have addressed my previous concerns and I have no further comments. I believe the revised manuscript is still of high quality and worth acceptance.

END OF COMMENTS

1st Confidential Review

22-Feb-2023

Response to Reviewing Editor:

- 1. In addition, thank you for making most of the p values exact. However, there are still some that are shown as < or >, and are not describing the results of multiple tests (e.g. In addition, baseline MEP latencies differed between coil orientations ($F_{1,156} = 247.41$, $P < 0.05$), with shorter PA-LM latencies compared to AP-LM latencies ($EMD = 1.9$ ms [1.6, 2.1], $P < 0.05$).). P-values will need to be given precisely throughout before the paper can be accepted for publication.*

Precise P -values are stated in the manuscript to three significant figures. P -values smaller than 0.0001 are stated as $P < 0.0001$, whereas P -values are stated as < or > 0.05 where multiple comparisons are covered by a single statement.

- 2. In addition, while it is good to see all the data points on the figures, the error bars are not described - I presume these are SD, but this is not stated. Please could the authors clarify for each figure.*

The data in the figures are presented as estimated marginal means (EMM) with 95% confidence intervals (95% CI), whereas the individual data points are the individual subject means. This has now been stated for each figure.

Response to Senior Editor:

- 1. Please ensure that the basis of the calculation of errors bars is stated in all instances (and that this is bodied in standard deviations, unless there are grounds for showing confidence intervals). Please also see the note from the Reviewing Editor concerning the reporting of P-values.*

We have now stated mean \pm standard deviation or EMM [95% CI] in all instances where appropriate throughout the manuscript and addressed the point from the Reviewing Editor regarding the reporting of P -values.

Dear Dr Liao,

Re: JP-RP-2023-284204R2 "Modulation of dorsal premotor cortex differentially influences I-wave excitability in primary motor cortex of young and older adults" by Wei-Yeh Liao, George M. Opie, Ulf Ziemann, and John G. Semmler

We are pleased to tell you that your paper has been accepted for publication in The Journal of Physiology.

Authors should note that it is too late at this point to offer corrections prior to proofing. The accepted version will be published online, ahead of the copy edited and typeset version being made available. Major corrections at proof stage, such as changes to figures, will be referred to the Editors for approval before they can be incorporated. Only minor changes, such as to style and consistency, should be made at proof stage. Changes that need to be made after proof stage will usually require a formal correction notice.

Yours sincerely,

Richard Carson
Senior Editor
The Journal of Physiology

P.S. - You can help your research get the attention it deserves! Check out Wiley's free Promotion Guide for best-practice recommendations for promoting your work at www.wileyauthors.com/eeo/guide. You can learn more about Wiley Editing Services which offers professional video, design, and writing services to create shareable video abstracts, infographics, conference posters, lay summaries, and research news stories for your research at www.wileyauthors.com/eeo/promotion.

IMPORTANT NOTICE ABOUT OPEN ACCESS: To assist authors whose funding agencies mandate public access to published research findings sooner than 12 months after publication, The Journal of Physiology allows authors to pay an Open Access (OA) fee to have their papers made freely available immediately on publication.

You can check if your funder or institution has a Wiley Open Access Account here: <https://authorservices.wiley.com/author-resources/Journal-Authors/licensing-and-open-access/open-access/author-compliance-tool.html>.

EDITOR COMMENTS

Reviewing Editor:

Thank you for addressing the remaining concerns in your resubmission.